# Functional Dentition and 12-Month Changes in Body Measurements among Thai Older Adults

**DOI:** 10.3390/ijerph17124200

**Published:** 2020-06-12

**Authors:** Piyada Gaewkhiew, Wael Sabbah, Eduardo Bernabé

**Affiliations:** 1Faculty of Dentistry, Oral & Craniofacial Sciences, King’s College London Dental Institute at Guy’s, King’s College and St. Thomas’ Hospitals, London SE5 9RS, UK; wael.sabbah@kcl.ac.uk (W.S.); eduardo.bernabe@kcl.ac.uk (E.B.); 2Department of Community Dentistry, Faculty of Dentistry, Mahidol University, Bangkok 10400, Thailand

**Keywords:** dentition, body weight and measures, aged, diet, longitudinal studies

## Abstract

This study evaluated the association of functional dentition with 12-month changes in body measurements and nutrient intake among older adults. Data from 651 community dwellers, aged 60 years and over, in Phetchaburi, Thailand, were analysed (retention rate: 83%). Data were collected via interviews (including a semi-structured food frequency questionnaire), anthropometric measurements and dental examinations. Associations were tested in linear regression models adjusted for baseline sociodemographic factors, behaviours, chronic conditions and medications. On average, participants experienced a significant increase in body mass index (BMI) and significant decreases in waist circumference (WC) and triceps skinfold thickness (TSF). A negative, albeit not significant, association between functional dentition and change in BMI was observed after adjusting for confounders. Whilst participants who had non-functional dentition without dentures experienced increases in BMI (predicted mean change: 0.25; 95% Confidence Interval: 0.09, 0.41), those who had non-functional dentition with dentures (0.21; 95%CI: −0.08, 0.50) and functional dentition (−0.07; 95%CI: −0.42, 0.28) remained stable. No similar trends were noted for WC or TSF. Functional dentition was not associated with changes in nutrient intake either. The findings provide little evidence on the association of functional dentition with short-term changes in nutrient intake or nutritional status.

## 1. Introduction

Tooth loss is a major oral health problem among older adults globally [1]. It is the complex result of oral diseases (such as dental caries and periodontal disease), access to dental care services and the individuals’ and dentists’ treatment preferences [2]. Incremental tooth loss could lead to a dentition that is no longer fit-for-purpose [3]. Evidence from systematic reviews of observational (mostly cross-sectional) studies suggests that having a full dentition is associated with better diet and nutritional status [4,5]. However, a recent systematic review of longitudinal studies reported inconclusive findings [6]. For nutrient intake, the only consistent association was for (self-reported) tooth loss and dietary cholesterol [7,8,9]. For nutritional status, two studies showed that tooth loss was associated with weight loss [10,11], one study with weight gain [12] and one reported no association [13]. Two recent longitudinal studies have added further contradicting evidence. On one hand, Kiesswetter, et al. [14] used two-wave panel data over a 10-year period and found no association between self-reported edentulism and body weight. On the other hand, Logan, et al. [15] reported that men with 21 or more teeth had higher intake of fruits, vegetables and nuts (measured once in the future only, not as change from baseline) than those with fewer than 21 teeth and dentures 13 years later.

There is a need for further cohort studies that include a clinical assessment of functional dentition (FD), which is a more robust measure of masticatory function than the simple count of remaining teeth [2]. A FD is based on the shortened dental arch (SDA) concept, which requires six anterior occluding pairs and at least four posterior occluding pairs [16]. Only one previous longitudinal study in this area has assessed FD with the SDA [17]. Furthermore, it has been noted that teeth need to be in good clinical condition to be functional because toothache and tooth mobility could affect function even when there is an opposing tooth [18]. Classifications of the dentition based on this notion have been recently proposed [19,20], but no single longitudinal study has evaluated them in relation to diet and nutritional status.

Most evidence on this area comes from developed countries [6]. New longitudinal studies in other countries are currently required. Settings where traditional diets, as opposed to Western-style diets, are still in place today would be relevant as diets continue to be based on starchy staples and fibre-rich foods. The Thai cuisine is internationally known for its blend of slow-cooked meat, steamed vegetables which could be seen in, for example, tom-yam, curries, or side dishes with chili pastes. Therefore, the Thai setting offers a unique opportunity to assess the role of FD in a non-Western diet. The purpose of this study was to determine the association between FD and 12-month changes in body measurements and nutrient intake among Thai older adults. It was hypothesised that older adults with FD would have better nutrient intake and body measurements over time than edentulous individuals.

## 2. Materials and Methods

This report is based on a 12-month longitudinal study carried out in Phetchaburi province, Thailand. At baseline, health authorities in four of the eight districts in the province agreed to be involved in the survey. In these four districts, six out of 16 urban and 11 out of 39 rural sub-districts agreed to take part in this study. A total of 788 free-living Thai adults, aged 60 years and over, were recruited for the baseline survey (response rate: 97%). All participants had normal scores in the abbreviated mental test score (>8 out of 10) [21] and in the activities of daily living questionnaire (≥12 out of 20) [22]. Participants who could not communicate or had disabling conditions were excluded [23]. All participants were invited to a follow-up examination a year later. Overall, 651 participants were re-examined (retention rate: 82.6%). The mean follow-up time was 11.6 months (SD: 0.8, range: 10 to 13).

All subjects provided written informed consent before participation in the study. The study protocol was approved by the Ethics Committee by the Research Ethics Committee of King’s College London (HR-17/18-4631) and Phetchaburi Health Authority (COA No.001/2561).

### 2.1. Data Collection

The baseline survey (2018) collected data via questionnaires, anthropometric measurements and dental examinations. The same methods were used in the follow-up survey (2019) except for dental examinations which were not undertaken. The questionnaire collected information on demographic characteristics (sex, age, marital status and residence area), socioeconomic conditions (education and wealth index), behaviours (smoking status and physical activity) and chronic conditions. The wealth index was constructed based on ownership of 10 household assets: air conditioner, bed, car, electric water boiler, flushing toilet, house telephone, kettle, microwave, personal computer and washing machine. Principal component analysis was used to extract the first common factor, which was subsequently used to obtain the wealth score and group it into quartiles [24]. Smoking status was determined via three questions [25]. Current smokers were those who reported smoking nowadays, and daily or less than daily; former smokers were those who reported smoking daily or less than daily in the past but not currently; and never smokers were those who did not fall in any of the two previous categories. Physical activity was calculated based on responses to four items on weekly frequency of strenuous, moderate, mild and walking exercise. Responses were used to derive a weighted measure of overall metabolically adjusted exercise-related physical activity, calculated as 2 × strenuous + moderate + mild + walking exercise sessions, following previous studies in Thailand [26,27]. The score was subsequently categorised into three groups (0–6, 7 and 8+ sessions/week). Participants were asked whether they have ever been diagnosed with hypertension, hyperlipidaemia and diabetes which are the three most common chronic conditions among older adults in Thailand [28]. If a condition was reported, participants were asked to report any medications taken.

The questionnaire also included a 154-item semi-quantitative Food Frequency Questionnaire (FFQ) to assess intake of food and beverages in the last month. The FFQ was specifically developed for Thai older adults, with good validity and reliability [29,30]. This FFQ covered 15 main dietary groups: (1) rice/carbohydrate-based dishes and ready meals, (2) beans and tofu products, (3) fresh and processed vegetables, (4) fresh and processed fruits, (5) meat and meat products, (6) fish and shellfish, (7) egg and egg products, (8) insects, (9) dairy products, (10) beverages and drinks, (11) sweet snacks and desserts, (12) snacks containing legumes, beans and seeds products, (13) sauces and seasonings, (14) chili paste and other pastes, and (15) Thai curry and soups. Participants reported their frequency of consumption of each food item using 7-point ordinal scales (from never to 3 or more times a day), and portion size using 7-point ordinal scales (from a quarter of a portion to 2 standard portions). Participants were assisted to estimate portion sizes using photos of cooking measurements (plates, cups, spoons, etc.). A standard portion size was specified in household measurements and grams for each food item. Daily intake of nutrients was calculated as daily servings multiplied by their nutrient content, which was summed across all food items. Nutrient intake was estimated using Thai food composition tables with INMUCAL V4.0 nutrient analysis software. The following nutrients were included in the present analysis: total energy intake (TEI), macronutrients (carbohydrate, protein, fat, saturated fat), cholesterol, dietary fibre, minerals (calcium and iron) and vitamins (A, B1, B2, B3, B6, B12, C and E). Thirty FFQs were repeated to assess test–retest reliability. The intra-class correlation values ranged between 0.75 and 0.97 for TEI and macronutrients.

Trained staff took all body measurements. Participants’ height (without shoes) was determined with a portable, free-standing stadiometer. Participants’ weight was determined with an electronic portable scale. Both were used to calculate participants’ body mass index (BMI). Waist circumference (WC) was measured at the end of normal respiration, at the midpoint between the lower part of the last rib and the top of the hip by measuring tape. Triceps skinfold thickness (TSF) was measured at the midpoint of the back upper arm with Harpenden callipers. All body measurements were performed in duplicate, and the average of both recordings was used for analysis. The primary outcome for this study was the absolute change in BMI, which was estimated by subtracting the baseline from the 12-month follow-up value. Secondary outcomes were 12-month changes in WC and TSF.

Dental examinations were conducted to determine the number of functional tooth units (FTU), the condition of teeth and dentures (if available). Two trained dentists performed all dental examinations on mobile dental chairs, under artificial light and using a mouth mirror and periodontal probe. A FTU was recorded as present if there were opposing teeth (natural or replaced with fixed prostheses). Opposing molars counted as two FTU whereas all other teeth counted as one FTU. Dental caries were diagnosed according to the World Health Organization (WHO) criteria [31]. Only severe tooth mobility was recorded, which was defined as Grade III (>2 mm of buccolingual or any vertical movement) according to the Grace and Smales [32] mobility index. Removable complete and partial dentures were recorded as present if they were worn on the examination day and while eating. Individuals were classified as having non-FD without dentures (reference group), non-FD with dentures or FD (i.e., having all 6 anterior FTU plus at least 4 posterior FTU, not necessarily symmetrical). If either tooth in a FTU had evidence of severe caries (root remnants) or severe mobility, the FTU was not counted as part of the requirement for a FD [23]. Seventy-eight participants were re-examined for reliability assessment. The inter- and intra-examiner Kappa values for FTU were 0.87 (95% CI: 0.83, 0.91) and 0.94 (95% CI: 0.86, 1.00), respectively.

### 2.2. Statistical Analysis

Data analysis was conducted in Stata SE version 15 (StataCorp LP, College Station, TX, USA). First, the baseline sociodemographic, behavioural and health-related characteristics of participants retained and lost to follow-up were compared using the Chi-squared test to assess the impact of attrition on the results. Next, the three FD groups (non-FD without dentures, non-FD with dentures and FD) were compared in terms of their baseline characteristics using the Chi-squared test. Thereafter, changes in BMI, WC and TSF were compared according to baseline characteristics. Student’s t-test was used when comparing two groups (sex, marital status, education and residence area) and analysis of variance (ANOVA) was used when comparing 3 or more groups (age, wealth quartiles, physical activity, smoking status, hypertension, hyperlipidaemia, diabetes and medications). Normality in the distribution of BMI, WC and TSF changes was tested using the Shapiro–Wilks test.

The association of functional dentition with 12-month change in BMI was tested in linear regression models adjusted for baseline confounders (sociodemographic factors, behaviours, chronic conditions, medication taken, and 12-month change in TEI). Unstandardized regression coefficients with 95% confidence intervals (CI) were therefore reported as the measure of association. Linear trends in the 12-month change in BMI according to FD groups were tested by fitting FD as a continuous, rather than as a categorical, variable in the regression model. The same modelling strategy was used for testing the associations of FD with 12-month changes in WC and TSF. Finally, the associations of FD with 12-month changes in TEI and every macronutrient were tested in linear regression models adjusted for confounders. Linear trends in the 12-month change in intake of each nutrient were tested by fitting FD as a continuous, rather than as a categorical, variable in the corresponding regression model.

## 3. Results

Participants lost to follow-up were more likely to be men, from rural areas, smokers and non-diabetic than those retained in the sample. In all, 14.1% of participants had FD, 20.1% had non-FD with dentures and 65.8% had non-FD without dentures. The comparison of baseline characteristics between participants with FD, non-FD with dentures and non-FD without dentures is shown in Table 1. Participants in the FD group were significantly younger and took more medications than those in the other two groups. No other differences between FD groups were noted.

Whilst there was a significant increase in BMI (0.20 ± 1.60 kg/m^2^) from baseline to follow-up, significant decreases in WC (−0.61 ± 5.27 cm) and TSF (−6.03 ± 9.14 mm) were also noted. Changes in each body measurement according to participants’ baseline characteristics are reported in Table 2. Significant differences in BMI change were only found between participants with and without hypertension. Those with controlled and uncontrolled hypertension had a significantly smaller increase in BMI than those without such chronic conditions. No differences in WC change were found according to covariates. Most differences by covariates were found in TSF. Female, younger, wealthier and less physically active participants, never smokers and those with controlled diabetes had larger reductions in TSF than their corresponding counterparts.

Table 3 reports the association between FD and 12-month change in body measurements. FD was negatively (albeit not significantly) associated with changes in BMI. Along an average increase in BMI over time observed in the full sample, participants who had FD showed non−significant increases in BMI compared to those who had non-FD with dentures and non-FD without dentures. The predicted mean BMI change was 0.25 (95%CI: 0.09, 0.41, *p* = 0.002) among participants who had non-FD without dentures, 0.21 (95%CI: −0.08, 0.50, *p* = 0.163) among those who had non-FD with dentures and −0.07 (95%CI: −0.42, 0.28, *p* = 0.689) among those who had FD. No similar trends were noted for change in WC or TSF.

There were significant increases in TEI as well as in intake of carbohydrates, sugars, dietary fibre, vitamins A, B1, B6, C and E, iron and calcium over time while there were decreases in intake of fat, dietary cholesterol and vitamin B12. No significant changes in the intake of protein and vitamins B2 and B3 were noted. No significant associations between FD and changes in TEI or intake of macronutrients and micronutrients were identified after adjustment for confounders (Table 4).

## 4. Discussion

This longitudinal study provided little support for the association of baseline FD with changes in body measurements among Thai older adults. Although having a FD was negatively associated with 12-month change in BMI, this association was not significant. In addition, having a FD was not associated with 12-month changes in other body measurements or changes in nutrient intake.

Although the association between FD and BMI change was not significant, the magnitude of the effect was such that while participants who had non-FD without dentures experienced a BMI increase (0.25 kg/m^2^), the BMI of participants who had non-FD with dentures and FD remained stable over the same period (0.21 and −0.07 kg/m^2^, respectively). Two possible explanations have been advanced in the literature for the association between tooth loss and nutritional status [13,14]. On one side, individuals with fewer remaining teeth would avoid hard-to-chew (fibrous or raw) foods due to their limited chewing ability [33], which could lead to inadequate nutrient intake and weight loss [10,34]. What is more likely though is that individuals with fewer remaining teeth would have greater consumption of soft-to-chew foods to compensate, including energy-dense food (rich in carbohydrates and/or fat), which subsequently could lead to greater chances of gaining weight [7,8,9,12]. Although not significant, the present finding suggests that having a FD would help older adults maintain their body composition. The same non-significant trend favouring older adults with FD was not observed with WC and TSF, which were chosen as indicators of abdominal and peripheral fat, respectively [35]. The increase in BMI along with decreases in WC and TSF observed in the sample could be explained by accumulation of fat in other parts of the body, such as thighs, legs and breasts [36], which are common body fat deposits among women (72.7% of our sample) that are not captured by WC or TSF.

The lack of association between baseline FD and changes in nutrient intake could be explained by participants’ engagement in adaptive as opposed to maladaptive eating behaviour [37]. Adaptive behaviours are those used to compensate for difficulties in chewing and allow the intake of a varied diet. They include modification in cooking method (cooking food until softened, chopping, mashing, peeling, shredding or grounding), food texture selection (adding a product, such as gravy, mayonnaise or butter, to moisten food or choosing softer foods), meal timing (taking more time to eat or eating before going out) and approaches to chewing (using the side of the mouth with more teeth) [37,38]. Maladaptive behaviours include avoiding fibre-rich and nutrient-dense foods that are conventionally difficult to chew (such as grains, nuts and seeds, read meats, raw vegetables and apples) as well as increasing the consumption of foods high in fats and sugars and other carbohydrates sources such as mashed potatoes and ice creams [33,37,38]. It is thus possible that older adults with non-FD could still reach their nutrient requirements by altering their eating experience.

In terms of practice and policy, the present findings suggest that having a FD might not be enough to improve nutrient intake and subsequent nutritional status among older adults. This is in line with clinical trials showing that combining the fitting of a new set of removable complete dentures with dietary advice or counselling might lead to improvements in dietary intake, at least in the short-term [39,40]. Therefore, the provision of prosthodontic care to older adults or rehabilitating patients to a FD should be combined with nutrition counselling and food modification training. From a research point of view, further evidence from prospective designs should include longer follow-up times, multiple assessments (more than two) of dietary intake and body measurements. A valuable addition would be to have multiple dental assessments to capture changes in FD (e.g., moving from a FD to a non-FD, becoming edentulous, or fitting a set of dentures) on diet and nutritional status. Finally, further studies should be preferably conducted in low-and-middle income countries, to help balance the evidence between Western and non-Western diets.

Some study limitations need to be discussed. First, the present findings provide evidence of associations rather than causal relationships. Second, selection bias could have arisen from two sources, sample selection and attrition. Although the response rate was high, the responses at sub-district and province levels were moderate only, which could explain the differences found against the wider population. Therefore, the present results cannot be generalised beyond the study sample. In addition, the retained sample included more females, participants from municipal areas, never smokers and diabetics than those lost to follow-up. Such differences imply that the findings discussed in previous sections should not be extrapolated beyond the study sample. Third, we did not collect information on how long participants had a non-FD, which could have provided valuable information in terms of duration of exposure and adaptation to their condition. Fourth, even when a validated semi-structured FFQ with a short recall time (previous month) was used for dietary assessment, responses relied on participants’ memory, judgements on portion size and frequency. Responses could also be affected by seasonal variation as some fruits are only consumed in the Summer such as mango, durian, rambutan, longan, custard apple, mangosteen, and lychee. In addition, a missing element in the semi-structured FFQ was the cooking method, which could impact on the estimation of nutrients due to cooking and increased temperature.

## 5. Conclusions

This 12-month longitudinal study among Thai older adults provided little evidence on the association between functional dentition and short-term changes in nutrient intake and nutritional status. Although a negative, non-significant trend in 12-month BMI change was observed across functional dentition groups, such trend was not replicated for changes in WC and TSF. Functional dentition was not associated with short-term changes in nutrient intake either.

## Figures and Tables

**Table 1 ijerph-17-04200-t001:** Comparison of baseline characteristics between participants who had non-functional functional dentition (FD) without dentures, non-FD with dentures and FD (*n* = 651).

Baseline Characteristics		All Sample		Non-FD without Dentures		Non-FD with Dentures		FD		*p* Value ^a^
		*n*	%	*n*	%	*n*	%	*n*	%	
Sex										0.212
	Men	178	27.3	125	29.2	28	21.4	25	27.2	
	Women	473	72.7	303	70.8	103	78.6	67	72.8	
Age groups										0.006
	60–64 years	201	30.9	128	29.9	33	25.2	40	43.5	
	65–69 years	147	22.6	96	22.4	24	18.3	27	29.3	
	70–74 years	123	18.9	78	18.2	33	25.2	12	13.0	
	75–79 years	92	14.1	65	15.2	19	14.5	8	8.7	
	80+ years	88	13.5	61	14.3	22	16.8	5	5.4	
Residence area										0.144
	Municipal	309	47.5	200	46.7	71	54.2	38	41.3	
	Non-municipal	342	52.5	228	53.5	60	45.8	54	58.7	
*Education*										0.460
	No education	83	12.7	54	12.6	14	10.7	15	16.3	
	Any education	568	87.3	374	87.4	117	89.3	77	83.7	
Wealth quartiles										0.438
	Q1 (poorest)	163	25.0	110	25.7	26	19.8	27	29.3	
	Q2	166	25.5	114	26.6	29	22.1	23	25.0	
	Q3	164	25.2	104	24.3	37	28.2	23	25.0	
	Q4 (wealthiest)	158	24.3	100	23.4	39	29.8	19	20.7	
Physical activity										0.380
	0–6 sessions/week	75	11.5	49	11.4	14	10.7	12	13.0	
	7 sessions/week	395	60.7	269	62.9	78	59.5	48	52.2	
	8+ sessions/week	181	27.8	110	25.7	39	29.8	32	34.8	
Smoking status										0.183
	Never	473	72.7	305	71.3	104	79.4	64	69.6	
	Former	125	19.2	82	19.2	21	16.0	22	23.9	
	Current	53	8.1	41	9.6	6	4.6	6	6.5	
Hypertension										0.144
	No	125	19.2	71	16.6	31	23.7	23	25.0	
	Yes, controlled	519	79.7	351	82.0	99	75.6	69	75.0	
	Yes, uncontrolled	7	1.1	6	1.4	1	0.8	0	0.0	
Hyperlipidaemia										0.491
	No	249	38.2	160	37.4	56	42.7	33	35.9	
	Yes, controlled	395	60.7	265	61.9	73	55.7	57	62.0	
	Yes, uncontrolled	7	1.1	3	0.7	2	1.5	2	2.2	
Diabetes										0.768
	No	421	64.7	272	63.6	91	69.5	58	63.0	
	Yes, controlled	225	34.6	153	35.7	39	29.8	33	35.9	
	Yes, uncontrolled	5	0.7	3	0.7	1	0.8	1	1.1	
Medications taken										0.005
	None taken	81	12.4	49	11.5	16	12.2	16	17.4	
	1–2 taken	284	43.6	181	42.3	74	56.5	29	31.5	
	3–4 taken	232	35.6	157	36.7	34	26.0	41	44.6	
	5+ taken	54	8.3	41	9.6	7	5.3	6	6.5	

^a^ Chi-squared test was used for comparison.

**Table 2 ijerph-17-04200-t002:** Mean change (SD) in body mass index (BMI), waist circumference (WC) and triceps skinfold thickness (TSF), by baseline characteristics of Thai older adults (*n* = 651).

Baseline Characteristics		BMI (kg/m^2^)	WC (cm)	TSF (mm)
All participants				
	Baseline value	24.51 ± 4.71	88.88 ± 10.97	25.48 ± 11.41
	Mean change	0.20 ± 1.60	−0.61 ± 5.27	−6.03 ± 9.14
Sex				
	Men	0.07 ± 1.60	−0.56 ± 5.36	−2.96 ± 7.19
	Women	0.25 ± 1.68	−0.63 ± 5.24	−7.19 ± 9.53
	*p* value ^a^	0.217	0.880	<0.001
Age groups				
	60–64 years	0.26 ± 1.59	−0.22 ± 5.05	−7.08 ± 8.77
	65–69 years	0.34 ± 1.55	−0.98 ± 5.20	−7.44 ± 9.90
	70–74 years	0.15 ± 1.56	−0.98 ± 5.98	−5.82 ± 9.76
	75–79 years	−0.07 ± 1.77	−0.98 ± 4.61	−4.84 ± 8.38
	80+ years	0.17 ± 1.98	0.04 ± 5.43	−2.83 ± 7.59
	*p* value	0.425	0.375	0.001
Residence area				
	Urban	0.15 ± 1.43	−0.53 ± 4.79	−5.76 ± 9.37
	Rural	0.24 ± 1.85	−0.68 ± 5.67	−6.28 ± 8.93
	*p* value	0.507	0.717	0.469
Education				
	No education	0.17 ± 1.72	−1.39 ± 5.46	−6.32 ± 9.44
	Any education	0.20 ± 1.66	−0.49 ± 5.23	−5.99 ± 9.10
	*p* value	0.871	0.149	0.760
Wealth quartiles				
	Q1 (poorest)	0.18 ± 1.92	−0.17 ± 5.31	−5.15 ± 10.18
	Q2	0.28 ± 1.63	−0.59 ± 4.81	−4.67 ± 8.21
	Q3	0.26 ± 1.41	−0.83 ± 5.31	−7.03 ± 8.74
	Q4 (wealthiest)	0.07 ± 1.65	−0.84 ± 5.64	−7.35 ± 9.11
	*p* value	0.654	0.635	0.015
Physical activity				
	0–6 sessions/week	0.23 ± 1.94	−0.60 ± 6.05	−7.95 ± 9.87
	7 sessions/week	0.22 ± 1.70	−0.74 ± 5.29	−6.34 ± 9.42
	8+ sessions/week	0.14 ± 1.44	−0.33 ± 4.89	−4.57 ± 7.96
	*p* value	0.872	0.689	0.015
Smoking status				
	Never	0.19 ± 1.73	−0.74 ± 5.30	−6.72 ± 9.42
	Former	0.20 ± 1.53	−0.42 ± 5.62	−3.75 ± 7.33
	Current	0.26 ± 1.31	0.12 ± 3.97	−5.27 ± 9.64
	*p* value	0.958	0.482	0.004
Hypertension				
	No	0.52 ± 1.75	−0.70 ± 4.79	−6.02 ± 8.84
	Yes, controlled	0.12 ± 1.64	−0.61 ± 5.39	−6.13 ± 9.24
	Yes, uncontrolled	0.40 ± 0.97	1.66 ± 3.66	0.48 ± 2.60
	*p* value	0.049	0.514	0.164
Hyperlipidaemia				
	No	0.27 ± 1.71	−0.30 ± 5.38	−5.54 ± 8.21
	Yes, controlled	0.16 ± 1.63	−0.76 ± 5.15	−6.36 ± 9.67
	Yes, uncontrolled	0.37 ± 2.17	−2.90 ± 7.31	−5.37 ± 9.72
	*p* value	0.462	0.292	0.528
Diabetes				
	No	0.28 ± 1.66	−0.25 ± 5.04	−5.14 ± 8.43
	Yes, controlled	0.06 ± 1.63	−1.30 ± 5.60	−7.76 ± 10.19
	Yes, uncontrolled	0.50 ± 2.90	0.4 ± 6.34	−3.80 ± 6.34
	*p* value	0.168	0.051	0.002
Medications taken				
	None taken	0.61 ± 1.78	−0.72 ± 5.05	−4.58 ± 7.40
	1–2 taken	0.14 ± 1.67	−0.34 ± 5.20	−6.15 ± 8.74
	3–4 taken	0.11 ± 1.69	−0.85 ± 5.34	−6.43 ± 9.80
	5+ taken	0.29 ± 1.18	−0.76 ± 5.70	−5.88 ± 10.51
	*p* value	0.104	0.729	0.468

^a^ Student’s *t*-test and ANOVA were used when comparing 2 and 3+ groups, respectively.

**Table 3 ijerph-17-04200-t003:** Models for the association between functional dentition (FD) and 12-month change in BMI, WC and TSF among Thai older adults (*n* = 651).

Functional Dentition	Mean	(SD)	Crude Associations		Adjusted Associations ^b^	
			Coef. ^a^	[95%CI]	Coef. ^a^	[95%CI]
Body mass index (kg/m^2^)						
Non-FD without dentures	0.25	(1.64)	0.00	[Reference]	0.00	[Reference]
Non-FD with dentures	0.17	(1.80)	−0.08	[−0.40, 0.25]	−0.05	[−0.38, 0.29]
FD	0.01	(1.54)	−0.24	[−0.62, 0.13]	−0.32	[−0.71, 0.06]
*p* value for trend ^c^			0.209		0.132	
Waist circumference (cm)						
Non-FD without dentures	−0.46	(5.25)	0.00	[Reference]	0.00	[Reference]
Non-FD with dentures	−0.88	(5.39)	−0.42	[−1.46, 0.61]	−0.33	[−1.39, 0.72]
FD	−0.88	(5.22)	−0.42	[−1.61, 0.77]	−0.28	[−1.50, 0.94]
*p* value for trend ^c^			0.377		0.535	
Triceps skinfold thickness (mm)						
Non-FD without dentures	−5.91	(9.29)	0.00	[Reference]	0.00	[Reference]
Non-FD with dentures	−6.74	(9.65)	−0.83	[−2.62, 0.96]	−0.56	[−2.31, 1.20]
FD	−5.62	(7.58)	0.29	[−1.78, 2.35]	0.95	[−1.07, 2.98]
*p* value for trend ^c^			0.924		0.576	

^a^ Linear regression was fitted and unstandardized regression coefficients (Coef.) reported. ^b^ Adjusted for gender, age in groups, residence area, education, wealth quartiles, smoking status, physical activity, hypertension, hyperlipidaemia, diabetes, medications taken, and 12-month change in total energy intake. ^c^ Linear trends were tested by fitting functional dentition as a continuous rather than as a categorical indicator.

**Table 4 ijerph-17-04200-t004:** Models for the association between FD and 12-month changes in total energy intake (TEI), macronutrients intake and micronutrients intake among Thai older adults (*n* = 651).

Nutrients	Non-FD with Dentures		FD		*p* Value for Trend ^b^
	Coef. ^a^	[95%CI]	Coef. ^a^	[95%CI]	
TEI (kcal)	128.54	[−219.60, 476.68]	−76.56	[−478.37, 325.26]	0.961
Protein (%TEI)	0.49	[−0.94, 1.94]	−0.37	[−2.04, 1.29]	0.901
CHO (%TEI)	−0.16	[−2.71, 2.38]	−0.25	[−3.19, 2.69]	0.848
Fat (%TEI)	−0.15	[−1.99, 1.69]	0.67	[−1.45, 2.80]	0.634
Cholesterol (mg)	22.05	[−41.83, 85.93]	−27.78	[−101.49, 45.92]	0.703
Dietary fibre (g)	0.87	[−2.43, 4.17]	0.87	[−2.94, 4.69]	0.562
Vitamin A (RAE/day)	7.43	[−64.69, 79.54]	−30.84	[−114.05, 52.36]	0.583
Vitamin B1 (mg/day)	−0.05	[−0.32, 0.22]	−0.23	[−0.55, 0.09]	0.170
Vitamin B2 (mg/day)	0.00	[−0.17, 0.18]	−0.15	[−0.35, 0.05]	0.209
Vitamin B3 (mg/day)	−1.40	[−4.19, 1.39]	−1.25	[−4.47, 1.97]	0.304
Vitamin B6 (mg/day)	−0.01	[−0.09, 0.07]	−0.02	[−0.11, 0.07]	0.602
Vitamin B12 (mcg/day)	0.02	[−0.15, 0.20]	−0.13	[−0.33, 0.07]	0.328
Vitamin C (mg/day)	0.22	[−6.98, 7.41]	5.57	[−2.73, 13.88]	0.248
Vitamin E (mg/day)	0.20	[−0.20, 0.60]	−0.12	[−0.58, 0.35]	0.955
Iron (mg/day)	−0.14	[−2.28, 2.00]	1.22	[−1.24, 3.69]	0.430
Calcium (mg/day)	−15.74	[−92.85, 61.36]	3.70	[−85.27, 92.67]	0.936

^a^ Linear regression was fitted and unstandardized regression coefficients (Coef.) reported. Models were adjusted for gender, age in groups, residence area, education, wealth quartiles, smoking status, physical activity, hypertension, hyperlipidaemia, diabetes, medications taken, and 12-month change in total energy intake (TEI) (except when modelling change in TEI). ^b^ Linear trends were tested by fitting functional dentition as a continuous rather as a categorical indicator. The reference group for comparison was non-FD without dentures (*n* = 428).

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
