# Peer review of "Functional Dentition and 12-Month Changes in Body Measurements among Thai Older Adults"

_ijerph, 2020, doi:10.3390/ijerph17124200_

Round 1

Reviewer 1 Report

My comments to the manuscript entitled „Functional Dentition and 12-Month Changes in Body  Measurements Among Thai Older Adults”:

  • The introduction requires major corrections, because it is prepared rather like a Discussion section. In the current version, it is not a good introduction for the presented research topic.
  • It is not known how did Authors verify the normality of distribution of data.
  • In Statistical analysis section Authors did not indicate that the ANOVA test was performed.
  • Table 2 is incomprehensible. It would be much better to provide mean values with a standard deviation rather than a mean change. In addition, detailed results for ANOVA should be indicated.
  • Authors state that they have conducted the study for 12 months, but they provided results only at the beginning and end of the study, so they did not observe the process of changes of the analyzed parameters within 12 months. It is not known how the parameters changed over 12 months, for example after 3, 6 months.
  • One sentence in the Conclusions section is definitely not enough.

In my opinion, most disturbing is that this manuscript is very similar to the authors' earlier article entitled “Functional dentition, dietary intake and nutritional status in Thai older adults.” (Gerodontology Pub Date : 2019-04-26 , DOI: 10.1111/ger.12408). Both experiments were conducted on 788 people, the introduction in both works was based on the exactly same articles (see references), the presentation of the data is the same (tables). In the first article Authors explored the relationship between functional dentition and nutritional status of Thai older adults. Authors conclude that “some evidence of an inverse association between functional dentition and being underweight, but not being overweight/obese, among Thai older adults” while in the present manuscript they analyzed changes of BMI at two time points, but without providing an average value, it is completely incomprehensible. In my opinion manuscript requires major corrections before being published.

Author Response

Comment: My comments to the manuscript entitled “Functional Dentition and 12-Month Changes in Body Measurements Among Thai Older Adults”:

The introduction requires major corrections, because it is prepared rather like a Discussion section. In the current version, it is not a good introduction for the presented research topic.

Response: We have removed our personal thoughts from the Introduction (1st and 2nd paragraph on page 2) and focussed on presenting available evidence and gaps in knowledge.

Comment: It is not known how did Authors verify the normality of distribution of data.

Response: We have added a sentence in Statistical Analysis (lines 147-148) to confirm that normality in the distribution of outcome measures was evaluated with the Shapiro-Wilks test.

Comment: In Statistical analysis section Authors did not indicate that the ANOVA test was performed.

Response: We now state in line 145 that ANOVA was used to compare 3 or more groups.

Comment: Table 2 is incomprehensible. It would be much better to provide mean values with a standard deviation rather than a mean change. In addition, detailed results for ANOVA should be indicated.

Response: The standard approach to analyse a numerical outcome measured twice (like in the present report) is to estimate change over time, what is also called the difference score (Singer and Willet, 2003; Clarke 2004; Aickin, 2009). To improve interpretability, we have added a line at the top of Table 2 to report baseline values for each body measurement. With this information, readers can judge how participants entered the study and how they changed after 12 months.

Singer JD, Willett JB. Applied Longitudinal Data Analysis: Modeling Change and Event Occurrence. Oxford University Press: Oxford; 2003.

Clarke PS. Causal Analysis of Individual Change Using the Difference Score. Epidemiology 2004;15(4):414-21.

Aickin M. Dealing with change: using the conditional change model for clinical research. Perm J 2009;13(2):80-4.

Comment: Authors state that they have conducted the study for 12 months, but they provided results only at the beginning and end of the study, so they did not observe the process of changes of the analyzed parameters within 12 months. It is not known how the parameters changed over 12 months, for example after 3, 6 months.

Response: We have replaced the term “over 12 months” with “12-month change” to avoid further confusion. We have also clarified in lines 120-121 that the outcome measure was estimated as the difference between the baseline and follow-up values for each body measurement.

Comment: One sentence in the Conclusions section is definitely not enough.

Response: The conclusion was expanded to describe all key findings of the study (lines 276-278).

Comment: In my opinion, most disturbing is that this manuscript is very similar to the authors' earlier article entitled “Functional dentition, dietary intake and nutritional status in Thai older adults.” (Gerodontology Pub Date : 2019-04-26 , DOI: 10.1111/ger.12408). Both experiments were conducted on 788 people, the introduction in both works was based on the exactly same articles (see references), the presentation of the data is the same (tables). In the first article Authors explored the relationship between functional dentition and nutritional status of Thai older adults. Authors conclude that “some evidence of an inverse association between functional dentition and being underweight, but not being overweight/obese, among Thai older adults” while in the present manuscript they analyzed changes of BMI at two time points, but without providing an average value, it is completely incomprehensible. In my opinion manuscript requires major corrections before being published.

Response: Our previous manuscript reported cross-sectional analyses of the baseline survey. Here we report the findings from the full study (longitudinal analysis of changes in body measurements). The sample is smaller (651 versus 788) due to losses to follow-up, but importantly the design is stronger than the cross-sectional design previously used, which might explain the different conclusions. We have defended our use of the change in body measures as the outcome in a previous comment.

Reviewer 2 Report

It is obvious there was a lot of effort and hard work behind this study. However, a lot of confounding factors have not been considered or studies. For example, medications and change in the medications that obviously have an impact on the patient's appetite, nutrition as well as weight loss or gain. 

The staus of the chronic diseases have not been considered either, well-controlled or poorly controlled hypertension or DM, which again has an impact on the wight change. 

It is suggested to go back and revised your information regarding the confounding factors 

Also, the duration of the edentulousisms or non-functional dentititon and changes in the body measure would be considered as it most likely happens closer to the date becoming edentulous or non-functional dentition. It might be the reason you did not see any association. You should consider adding that data too. 

Author Response

Comment: It is obvious there was a lot of effort and hard work behind this study. However, a lot of confounding factors have not been considered or studies. For example, medications and change in the medications that obviously have an impact on the patient's appetite, nutrition as well as weight loss or gain.

Response: Information on medication use was collected at baseline. This information has now been included in the analysis. The text of results and Tables 1 to 4 have been revised accordingly. Of note is that the conclusions of the study remained unchanged after these changes.  

Comment: The status of the chronic diseases have not been considered either, well-controlled or poorly controlled hypertension or DM, which again has an impact on the weight change. It is suggested to go back and revised your information regarding the confounding factors

Response: We now report each chronic condition in three categories: no; yes, controlled [receiving medication]; and yes, uncontrolled [without medication]. We reran all the analyses using these three indicators plus medications taken. The text of results and Tables 1 to 4 have been revised accordingly. Of note is that the conclusions of the study remained unchanged after these changes.

Comment: Also, the duration of the edentulousisms or non-functional dentititon and changes in the body measure would be considered as it most likely happens closer to the date becoming edentulous or non-functional dentition. It might be the reason you did not see any association. You should consider adding that data too.

Response: This information was not collected as part of the study. We have addressed this point in the discussion (lines 255-256).

Reviewer 3 Report

The manuscript is interesting.

The question which I have, why You didn't check the dental status after 12 months?

Also, caries can have a big influence on the quality of food - pain can influence the choice of the consistency of the food.

Author Response

Comment: The manuscript is interesting. The question which I have, why You didn't check the dental status after 12 months?

Response: Because of time constraints during the follow-up survey a compromise was made in terms of the information that could be collected without overburdening participants. We prioritised collecting data on body measurements and diet over dental data. This decision was taken assuming that few participants would experience major changes in dental status after just one year (i.e. moving from the functional to the non-functional dentition category). This point has been addressed in lines 252-254.

Comment: Also, caries can have a big influence on the quality of food - pain can influence the choice of the consistency of the food.

Response: When counting the number of functional occluding pairs, we excluded teeth with severe dental caries (and those with severe mobility) as both conditions could affect the function of the teeth in an occlusal contact. This is now clearly described in line 133.

Round 2

Reviewer 1 Report

In my opinion the introduction still requires corrections. Introductory sentences to the problem of tooth loss are needed.

Authors should use other articles to make this Introduction different from the previous article (“Functional dentition, dietary intake and nutritional status in Thai older adults.” (Gerodontology Pub Date : 2019-04-26 , DOI: 10.1111/ger.12408).

Author Response

Thank you for the opportunity to revise this paper for a second resubmission. We have modified the manuscript in accordance with the comments of the two reviewers. The attached document provides our point-by-point responses to these comments and describes the changes we have made to the text, which were highlighted on coloured text as requested.

Comment: In my opinion the introduction still requires corrections. Introductory sentences to the problem of tooth loss are needed.

Response: We have added a couple of sentences to the 1st paragraph of Introduction (lines 29-32) to describe the research problem (tooth loss).

Comment: Authors should use other articles to make this Introduction different from the previous article (“Functional dentition, dietary intake and nutritional status in Thai older adults.” (Gerodontology Pub Date : 2019-04-26 , DOI: 10.1111/ger.12408).

Response: We have updated our review of the literature (lines 37-42) and updated our list of references used in the introduction (highlighted in yellow).

Reviewer 2 Report

Thank you for addressing the comments. Although one of the main factors that is adaptation and duration of being edentulous or having non-functional dentition is not included, you do not gather that. So, just mention it in your discussion that you have not gathered this information in your study. 

Author Response

Thank you for the opportunity to revise this paper for a second resubmission. We have modified the manuscript in accordance with the comments of the two reviewers. The attached document provides our point-by-point responses to these comments and describes the changes we have made to the text, which were highlighted on coloured text as requested.

Comment: Thank you for addressing the comments. Although one of the main factors that is adaptation and duration of being edentulous or having non-functional dentition is not included, you do not gather that. So, just mention it in your discussion that you have not gathered this information in your study.

Response: We now address this point among the limitations of the study (lines 267-269).